

# Relationship between respiratory muscles ultrasound parameters and running tests performance in adolescent football players. A pilot study

Małgorzata Pałac[1,2], Damian Sikora[1], Tomasz Wolny[1,2] and Paweł Linek[1,2]

[1] Musculoskeletal Elastography and Ultrasonography Laboratory, Institute of Physiotherapy and Health Sciences, The Jerzy Kukuczka Academy of Physical Education, Katowice, Śląskie, Poland
[2] Musculoskeletal Diagnostic and Physiotherapy - Research Team, The Jerzy Kukuczka Academy of Physical Education, Katowice, Poland

## ABSTRACT

**Purpose**. Assessing the relationship between ultrasound imaging of respiratory muscles during tidal breathing and running tests (endurance and speed) in adolescent football players.

**Methods**. Ultrasound parameters of the diaphragm and intercostal muscles (shear modulus, thickness, excursion, and velocity), speed (30-m distance), and endurance parameters (multi-stage 20-m shuttle run test) were measured in 22 male adolescent football players. The relation between ultrasound and running tests were analysed by Spearman's correlation.

**Results**. Diaphragm shear modulus at the end of tidal inspiration was moderately negatively ($R = -0.49; p = 0.2$) correlated with the speed score at 10 m. The diaphragm and intercostal muscle shear modulus ratio was moderately to strongly negatively correlated with the speed score at 10 m and 30 m (about $R = -0.48; p = 0.03$). Diaphragm excursion was positively correlated with the speed score at 5 m ($R = 0.46; p = 0.04$) and 10 m ($R = 0.52; p = 0.02$). Diaphragm velocity was moderately positively correlated with the speed score at 5 m ($R = 0.42; p = 0.06$) and 30 m ($R = 0.42; p = 0.07$). Ultrasound parameters were not significantly related to all endurance parameters ($R \leq 0.36; p \geq 0.11$).

**Conclusions**. Ultrasound parameters of the respiratory muscles are related to speed score in adolescent football players. The current state of knowledge does not allow us to clearly define how important the respiratory muscles' ultrasound parameters can be in predicting some performance parameters in adolescent athletes.

Corresponding author
Małgorzata Pałac,
malgorzatapalac3@gmail.com

## INTRODUCTION

It is well known that respiratory function is related to physical activity and affects exercise performance in athletes. Respiratory muscles (RMs) are an integral part of the respiratory system and physical activity. Their morphology and contractile properties make them useful in endurance types of training (*Welch, Kipp & Sheel, 2019*). RMs are susceptible to
fatigue, resulting in reduced performance (*Aliverti, 2016*; *Welch, Kipp & Sheel, 2019*) and insufficient oxygen supply to the working muscles (*Mcconnell & Lomax, 2006*). Studies have shown that RMs training improves RMs' parameters and decreases muscle fatigue, resulting in a change in respiratory system function (*Welch, Kipp & Sheel, 2019*). It is also indicated that inspiratory muscle training affects the test results involving time trials or exercise endurance time (*Hajghanbari et al., 2013*). The main RMs are the diaphragm (DA) and intercostal muscles (IMs). Physiologically, the DA executes about 65% of the respiratory work during inspiration (*Moeliono, DM & Nashrulloh, 2022*) and affects to a greater extent lung movements (*Welch, Kipp & Sheel, 2019*). IMs, in turn, contribute to chest expansion (*Yoshida et al., 2021*), leading to increased inspiratory volume (*Yoshida et al., 2019*). During inspiration, while the IMs contract, the abdominal muscles gradually relax, and vice versa during expiration. This mechanism has some effects: (a) it prevents rib cage distortion; (b) the DA is unloaded and can act as a flow generator; and (c) the abdominal volume decreases below resting levels (*Aliverti, 2016*).

In football, RM training improves RMs' strength, which helps to improve exercise tolerance and lower blood lactate levels (*Guy, Edwards & Deakin, 2014*). Respiration exercises also improve muscle oxygen supply during high-intensity exercise (*Archiza et al., 2018*). This process can be translated into an improvement in fatigue tolerance and running efficiency of football players (*Archiza et al., 2018*). Additionally, it was confirmed that in youth football players, the RMs improve aerobic endurance, which is one of the most important parameters of motor preparation in football (*Mackała et al., 2020*).

Spirometry, as a gold standard of assessing respiratory function (*Durmic et al., 2015*), allows reproducible and standardised assessment of pulmonary function (*Lazovic-Popovic et al., 2016*). However, spirometry performance is the result of many factors (including airway obstruction, respiratory compliance, and RM strength) that do not allow direct analysis of the RMs (*Pałac et al., 2022*). In contrast, ultrasound (US) imaging can directly and reliably assess the thickness, excursion, and shear modulus (elasticity) of the RMs (*Pałac et al., 2022* ; *Zhu et al., 2019*). *Pałac et al. (2022)* also confirmed the reliability of RMs US measurements in adolescent football players. In the literature, some studies have shown the relationship between US parameters of the RMs and spirometry parameters in different populations (*Pałac & Linek, 2022*). However, a recent systematic review by *Pałac & Linek (2022)* has shown that the relationship between US parameters and lung function (measured, for example, by spirometry) is inconclusive. Thus, the two methods of measurement should not be used interchangeably, as they measure different aspects (*Pałac & Linek, 2022*).

Taking into account that RMs training affects motor skills and has implications for sports training, it is worth considering these muscles in athletes. Running tests are usually used to assess motor skills such as speed and endurance. According to the literature, speed and endurance depend on the thickness of the lower-extremity muscles, which has been measured using US in young athletes (*Stock et al., 2017*). Other US parameters have been related to motor skills in elite sports (*Sarto et al., 2021*). For example, RMs function correlates with postural stability in footballers (*León-Morillas et al., 2021*), and thus potentially affects motor skills as well. To the best of our knowledge, however, there

have been no studies relating US measurements of RMs with motor skills (endurance and speed) in adolescent football players. We believe that such an analysis is justified, as it may launch the exploration of RMs US measurements that are potentially useful in predicting motor skill performance in athletes. The aim of this preliminary report was to assess the relationship between US of RMs during tidal breathing and selected motor skill (endurance and speed) performance in adolescent football players. Based on the current state of the art, we hypothesised that endurance and speed parameters should be related to the thickness and elasticity of RMs (DA and IMs) in adolescent football players.

## MATERIALS & METHODS

### Informed consent

The study was approved by the Ethics Committee of the Jerzy Kukuczka Academy of Physical Education in Katowice (Decision No. 9/2020) and conducted in accordance with the guidelines of the Declaration of Helsinki. Before the study, participants and their parents were informed about all procedures performed and have given written consent to participate. All participants provided written informed consent to participate in the study. This research did not receive any external funding.

### Setting and study design

US data were collected in a laboratory setting (Institute of Physiotherapy and Health Sciences, Musculoskeletal Elastography and Ultrasonography Laboratory) by two physiotherapists, whereas endurance and speed measurements were performed by a motor preparation assistant on a football field with an artificial ground surface. Speed and endurance tests were conducted during two consecutive training days. During the first day, speed tests were performed, and on the next day, an endurance test was conducted. All measurements were performed in a preparation phase for the next football season. Due to organisational issues, US was collected one week after the endurance measurements.

### Participants

Adolescent footballers from the professional football academy were considered for the study. We invited all male individuals from a randomly selected team (one age group). The basic criteria of eligibility for the study were (a) all players had to be free of any health or injury issues at the time of testing; (b) no respiratory-related medical history; and (c) no surgical procedure on the pectoral chest, abdominal cavity, pelvic girdle, and/or spine. Information regarding the athletes' health was obtained by a short interview with the footballers and a coach or physiotherapist working with these athletes in the club.

### Ultrasound measurements

All US measurements were collected by an Aixplorer US scanner (Product Version 12.2.0, Software Version 12.2.0.808; Supersonic Imagine, Aix-en-Provence, France). Linear transducer array (2–10 MHz; SuperLinear 10-2, Vermon, Tours, France) in the SWE mode was used to evaluate the shear modulus and thickness of the ICs and DA on the right side of the body. Each participant laid in the supine position with the right hand
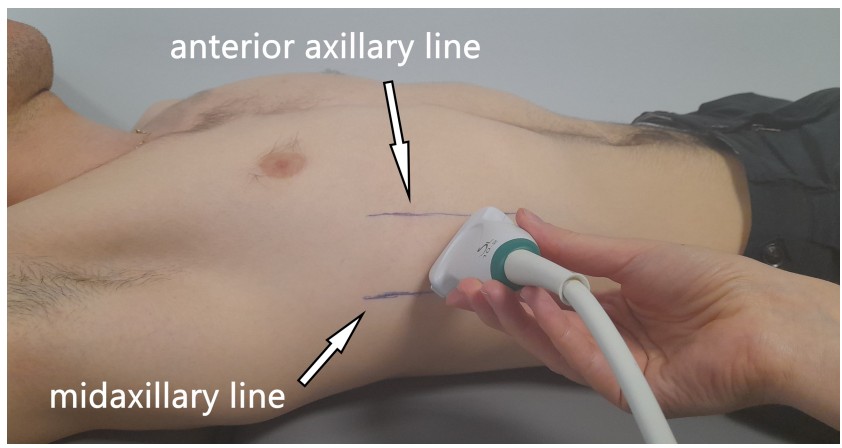

**Figure 1** Illustration showing the ultrasound probe placement and orientation (parallel to the ribs).

placed under the head in order to better visualise the DA. At the beginning, anterior and mid-axillary lines were marked on the participant's chest, and the US probe was positioned between the lines (Fig. 1). The probe was positioned in the first intercostal space (counting from the bottom) where the lungs did not obscure the DA during tidal breathing. The US measurements were performed in a longitudinal probe position (parallel to the ribs). The participants were asked to relax and breath quietly throughout the procedure. US data were collected twice at the end-tidal inspiration and at the end-tidal expiration, separately. The reliability of RM measurements has been confirmed in previous studies on healthy adolescent football players (*Pałac & Linek, 2022*).

DA excursion was collected in the M-mode on the Aixplorer US scanner coupled with convex transducer array (1–6 MHz, Cristal Curved XC6-1; Vermon, Tours, France). For the excursion measurement, the participant was in the supine position with the upper limbs along the trunk. The probe was placed in the right subcostal area. The participant was asked to take a maximal inspiration and then quietly expire. For the excursion DA measurement, a video collecting the work of breathing before maximal inspiration (tidal expiration) and during maximal inspiration and tidal expiration was recorded. The reliability of DA excursion was confirmed on athletes (*Calvo-Lobo et al., 2019*). DA excursion amplitude was described as the upright perpendicular distance from the minimum to the maximum point of DA displacement during a given breathing manoeuvre. DA excursion velocity is described as the velocity of DA displacement (during a given breathing pattern).

Shear modulus and thickness were calculated from the US images. The Q-Box™ quantitative tool was used to quantify muscle shear modulus. Three separate circles were positioned inside the fascial edge of each muscle, and the shear modulus was automatically calculated. The images were then saved on an external drive in DICOM format and transferred to a computer, where the muscle thickness was measured using RadiAnt DICOM Viewer (Medixant, Poznań, Poland). The DA thickness was measured between the pleural and peritoneal lines. The ICs were measured as the first more superficial

muscle than the DA. The thickness and shear modulus ratio was also measured as the end-inspiratory US value divided by the end-expiratory US value.

## Running tests

Two running tests were used to analyse the participants' endurance and speed. All measurements were collected by using photocells of the Witty System (Microgate Bolzano, Italy) with an accuracy of 0.01 s. The Witty System was coupled with Witty Manager (1.14.32 version; Microgate Bolzano, Italy) and connected to a laptop, allowing data collection (*Altmann et al., 2019*). Both tests were performed on a dry grass football pitch on a sunny day, and the participants wore football kit and boots.

Endurance was assessed by a progressive, multi-stage 20-m shuttle run test (MSRT) as a modification of the beep test (*Green et al., 2013*). The beep test requires athletes to run back and forth ("shuttle") between two cones separated by 20 m. The initial speed was 2.22 m/s for 1 min. At the end of the first min, the speed increased to 2.5 m/s and progressively increased by 0.14 m/s each min thereafter. The speed was imposed by audible beeps from pre-recorded audio. Each min stage (level) consisted of multiple "shuttles", and the number depended on the stage speed. Participants were advised to keep running at the pace of the beeps for as long as possible. Once the participant could no longer keep pace with the beeps (*i.e.,* failed to complete two consecutive shuttles in time), the test was terminated (*Green et al., 2013*). For the purpose of the study, we calculated the parameter "Total" as the total number of completed 20-m repetitions (during the whole test). The following parameters were used for further analysis: Total and calculated $VO_{2max}$ ($ml \cdot kg^{-1} \cdot min^{-1}$). $VO_{2max}$ was estimated from the maximal speed attained during the test *via* the previously developed prediction equation $-24.4 + 6.0 \times$ maximum aerobic speed (sec) (*Léger et al., 1988*).

The speed test involved running 30 m as fast as possible in a straight line between the photocells. Before the test began, the participants stood adjacent to (*i.e.,* their toes were not touching) the starting line in a standing split-stance position. They were instructed to run as fast as possible and slow down after crossing the finish line. A sound signal marked the beginning of each test. The timer was switched on when the starting line was passed, and measurements were automatically taken at 5 m, 10 m, and 30 m by the photocells positioned at those distances. The timer stopped when the finishing line was passed. Each participant ran the course twice, and the mean scores from both were analysed (*Altmann et al., 2019*).

## Statistical analysis

Data were analysed using Statistica 13.1 PL (Statsoft, Tulsa, OK, USA) and Excel (Microsoft Corporation, USA) software. Due to the non-normality of the distribution in the Shapiro–Wilk test, we decided to use Spearman's correlation in the analysis. The correlation value (R) was interpreted as follows: 0 to 0.30 or 0 to $-0.30$ was considered a weak correlation; 0.31 to 0.50 or $-0.31$ to $-0.50$ a moderate correlation; 0.51 to 0.70 or $-0.51$ to $-0.70$ a strong correlation; and 0.71 to 1 or $-0.71$ to $-1$ a very strong correlation (*Hopkins et al., 2009*). The significance level was set at $p \leq 0.05$. For the a priori analysis, the sample

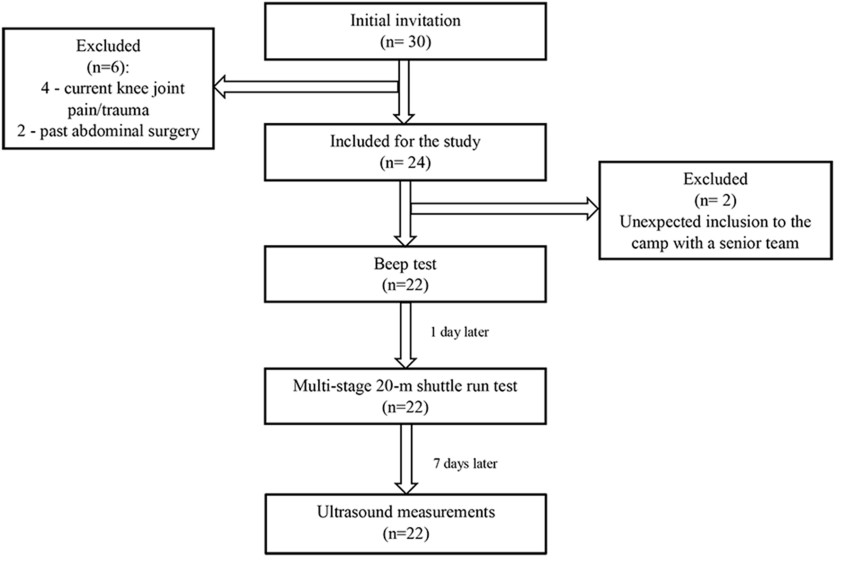

**Figure 2  Flow chart.**

size was determined using G*POWER (Version 3.1.9.7; Universität Kiel, Kiel, Germany) using an alpha of 0.05, a power of 0.80, and an effect size of 0.50 for a two-tailed test. Because Spearman's rank correlation coefficient is computationally identical to Pearson's product-moment coefficient, we used the software to calculate the latter.

# RESULTS

## Participants

Based on the assumptions, the required sample size was determined to be 26. Out of 30 initially invited footballers, 24 met the inclusion criteria. However, during the measurements, two athletes were at a camp with the senior team. Thus, a total of 22 adolescent footballers (two goalkeepers, eight defenders, nine midfielders, three forwards) were included in the final analysis (Fig. 2). Basic data and all parameters measured are shown in Table 1.

## Speed test *vs* US

DA shear modulus at the end of tidal inspiration was moderately negatively correlated with the speed score at 10 m. The DA shear modulus ratio was moderately negatively correlated with the speed score at 10 m and 30 m. The IC shear modulus ratio was moderately negatively correlated with the speed score at 10 m and strongly negatively correlated with the speed score at 30 m. Additionally, DA excursion was positively correlated with the speed score at 5 m (moderate) and 10 m (strong). DA velocity was moderately positively correlated with the speed score at 5 and 30 m, but statistical significance was borderline ($p = 0.06$). Detailed R values for each correlation are presented in Table 2.

**Table 1** Experimental group characteristics: anthropometric data, ultrasound parameters, endurance test (multi-stage 20-m shuttle run test), and speed test (straight line speed in 5, 10, and 30 m).

| Characteristic ($n = 22$) | mean ± SD | median |
|---|---|---|
| *Anthropometric data* | | |
| Age (yr) | 17.1 ± 0.29 | 17.0 |
| Body mass (kg) | 71.4 ± 7.74 | 70.0 |
| Body height (cm) | 180 ± 5.76 | 180 |
| BMI (kg/m$^2$) | 22.1 ± 1.95 | 22.0 |
| Football practice (yr) | 7.77 ± 0.75 | 8.0 |
| *SWE - Shear modulus (kPa)* | | |
| Diaphragm at the end of tidal inspiration | 31.2 ± 6.26 | 31.5 |
| Diaphragm at the end of tidal expiration | 29.4 ± 5.60 | 27.9 |
| Diaphragm ratio | 1.07 ± 0.18 | 1.05 |
| Intercostal muscle at the end of tidal inspiration | 27.1 ± 6.23 | 26.6 |
| Intercostal muscle at the end of tidal expiration | 27.0 ± 6.00 | 25.7 |
| Intercostal muscle ratio | 1.01 ± 0.15 | 0.97 |
| *B-mode – Thickness (mm)* | | |
| Diaphragm at the end of tidal inspiration | 2.09 ± 0.85 | 1.82 |
| Diaphragm at the end of tidal expiration | 1.71 ± 0.59 | 1.48 |
| Diaphragm ratio | 1.21 ± 0.21 | 1.20 |
| Intercostal muscle at the end of tidal inspiration | 3.98 ± 0.85 | 4.05 |
| Intercostal muscle at the end of tidal expiration | 4.09 ± 0.89 | 3.97 |
| Intercostal muscle ratio | 0.99 ± 0.15 | 0.95 |
| *M-mode* | | |
| Diaphragm excursion (cm) | 4.73 ± 1.45 | 4.59 |
| Diaphragm velocity (cm/s) | 2.13 ± 0.89 | 1.83 |
| *Multi stage 20-m shuttle run test* | | |
| Total | 127 ± 13.2 | 122 |
| calculated VO$_2$max (ml · kg$^{-1}$ · min$^{-1}$) | 56.2 ± 3.54 | 55.1 |
| *Speed test (s)* | | |
| Distance 5 m | 1.03 ± 0.05 | 1.03 |
| Distance 10 m | 1.87 ± 0.52 | 1.77 |
| Distance 30 m | 4.19 ± 0.20 | 4.14 |

**Notes.**
SD, standard deviation; BMI, Body Mass Index; SWE, shear wave elastography; ratio, diaphragm at the end of tidal inspiration/diaphragm at the end of tidal expiration; Total, total number of completed 20-m repetitions.

## MSRT *vs* US

US parameters were not significantly related to endurance parameters, although correlations varied from weak to moderate. Detailed R values for each correlation are presented in Table 3.

## DISCUSSION

The preliminary report was designed to assess the relationship between US of RMs during tidal breathing and selected motor skill (endurance and speed) performance in adolescent football players. To the best of our knowledge, there has not yet been a

**Table 2** Correlations between ultrasound parameters and speed test results.

| | 5 m | | 10 m | | 30 m | |
|---|---|---|---|---|---|---|
| | R | p | R | p | R | p |
| *Shear modulus* | | | | | | |
| Diaphragm at the end of tidal inspiration | −0.34 | 0.12 | −0.49 | 0.02[*] | −0.24 | 0.29 |
| Diaphragm at the end of tidal expiration | −0.10 | 0.66 | −0.14 | 0.55 | 0.10 | 0.66 |
| Diaphragm ratio | −0.31 | 0.16 | −0.48 | 0.02[*] | −0.41 | 0.06 |
| Intercostal muscle at the end of tidal inspiration | −0.26 | 0.26 | −0.39 | 0.08 | −0.18 | 0.44 |
| Intercostal muscle at the end of tidal expiration | −0.13 | 0.58 | −0.16 | 0.49 | 0.16 | 0.48 |
| Intercostal muscle ratio | −0.28 | 0.22 | −0.47 | 0.03[*] | −0.54 | 0.01[*] |
| *Thickness* | | | | | | |
| Diaphragm at the end of tidal inspiration | −0.07 | 0.75 | −0.06 | 0.80 | 0.22 | 0.34 |
| Diaphragm at the end of tidal expiration | −0.27 | 0.23 | −0.12 | 0.60 | 0.25 | 0.25 |
| Diaphragm ratio | 0.33 | 0.13 | 0.07 | 0.75 | −0.03 | 0.91 |
| Intercostal muscle at the end of tidal inspiration | −0.19 | 0.42 | −0.07 | 0.78 | 0.11 | 0.63 |
| Intercostal muscle at the end of tidal expiration | −0.08 | 0.74 | −0.11 | 0.64 | 0.05 | 0.83 |
| Intercostal muscle ratio | −0.04 | 0.86 | 0.14 | 0.56 | 0.07 | 0.76 |
| *M-mode* | | | | | | |
| Diaphragm excursion | 0.46 | 0.04[*] | 0.52 | 0.02[*] | 0.26 | 0.27 |
| Diaphragm velocity | 0.42 | 0.06 | 0.34 | 0.15 | 0.42 | 0.07 |

**Notes.**
SWE, shear wave elastography.
*statistically significant $p < 0.05$.
R, correlation coefficient; p, probability value; ratio, diaphragm at the end of tidal inspiration/diaphragm at the end of tidal expiration.

study relating the shear modulus, thickness, excursion, and velocity of the DA and ICs with parameters of speed and aerobic endurance based on MSRT in adolescent football players. This preliminary study has shown that US of RMs measurements (shear modulus, thickness, excursion, velocity) corresponded to speed in adolescent athletes. Thus, our initial hypothesis was partially confirmed because footballers with higher values of DA shear modulus at the end of tidal inspiration obtained better results in the 10-m speed test. Similarly, a higher DA and IC shear modulus ratio corresponded to a better speed score at 10 and 30 m, and a higher value of DA excursion and velocity was related to worse scores during the speed test. In turn, our results rejected the hypothesis that RMs are related to endurance in adolescent footballers.

## Speed

Taking all the results together, our study shows that RM shear modulus during tidal breathing may be partially related to the speed score in adolescent footballers. The shear modulus value is related to passive muscle force (*Koo & Hug, 2015*) and can be used to estimate changes in muscle force (*Ateş et al., 2015*). *Chino et al. (2018)* showed that DA shear modulus is non-linearly related to inspiratory mouth pressure, increasing rapidly at low inspiratory mouth pressure levels, but less rapidly as mouth pressure reaches higher levels. It can therefore be stated that a higher value of the DA shear modulus indicates greater inspiratory muscle strength. Another study confirmed that DA stiffness increases during

**Table 3  Relationship between ultrasound parameters and endurance test (multi-stage 20-m shuttle run) results.**

| | Total | | VO$_2$max | |
|---|---|---|---|---|
| | **R** | **p** | **R** | **p** |
| *Shear modulus* | | | | |
| Diaphragm at the end of tidal inspiration | −0.16 | 0.49 | 0.07 | 0.76 |
| Diaphragm at the end of tidal expiration | −0.17 | 0.46 | −0.05 | 0.83 |
| Diaphragm ratio | 0.03 | 0.88 | 0.18 | 0.42 |
| Intercostal muscle at the end of tidal inspiration | 0.03 | 0.90 | 0.33 | 0.15 |
| Intercostal muscle at the end of tidal expiration | −0.05 | 0.84 | 0.17 | 0.47 |
| Intercostal muscle ratio | 0.18 | 0.43 | 0.32 | 0.16 |
| *Thickness* | | | | |
| Diaphragm at the end of tidal inspiration | 0.01 | 0.98 | 0.17 | 0.45 |
| Diaphragm at the end of tidal expiration | 0.10 | 0.66 | 0.29 | 0.18 |
| Diaphragm ratio | −0.14 | 0.53 | −0.12 | 0.59 |
| Intercostal muscle at the end of tidal inspiration | 0.20 | 0.40 | 0.36 | 0.11 |
| Intercostal muscle at the end of tidal expiration | −0.01 | 0.97 | 0.10 | 0.66 |
| Intercostal muscle ratio | 0.16 | 0.48 | 0.25 | 0.27 |
| *M-mode* | | | | |
| Diaphragm excursion | 0.19 | 0.41 | −0.03 | 0.91 |
| Diaphragm velocity | 0.20 | 0.41 | 0.17 | 0.49 |

**Notes.**
SWE, shear wave elastography; Total, total number of completed 20-m repetitions; VO$_2$max, calculated VO$_2$max (ml · kg$^{-1}$ · min$^{-1}$); R, correlation coefficient; p, probability value; ratio, diaphragm at the end of tidal inspiration/diaphragm at the end of tidal expiration.

inspiration (*Şendur et al., 2022*). Our study shows that a stiffer (higher shear modulus value) DA during tidal inspiration characterised athletes with a better score in the speed test. This may indicate that a stiffer DA improves speed performance.

The DA shear modulus value is also related to transdiaphragmatic pressure (*Bachasson et al., 2018*), which is considered the gold standard for DA examination (*Ricoy et al., 2019*). Transdiaphragmatic pressure is the main measurement for determining DA strength (*Hamnegard et al., 1995*) and is clinically relevant because it represents the actual force that drives changes in lung volume and therefore ultimately alveolar ventilation (*Bachasson et al., 2018*). Sprint running (up to 6 s/up to 40 m) is characterised by anaerobic effort (*Sanders et al., 2017*). In our study, therefore, it can be assumed that the athletes had an anaerobic effort at the 30-m distance, so they were running at apnoea. It has been suggested that there is increased chest pressure during the initial phase of the speed test, which is linked to the Valsalva test (*Turban, 2010*). The Valsalva manoeuvre initiates with deep inhalation and DA downward movement (*Talasz et al., 2012*). Thus, the DA seems to be the main muscle involved in the Valsalva manoeuvre. The increased DA shear modulus during tidal breathing may predispose to a stronger DA contraction during the speed trial, resulting in a better score in the initial phase of running.

At a distance of 30 m, the DA and IC shear modulus ratio seems to be more significant. The ratio is calculated by dividing the shear modulus value at the peak of tidal inspiration

by the shear modulus value at the peak of tidal expiration. In our study, the higher the DA and IC shear modulus ratio, the better the speed test score. The ratio score is therefore determined not only by the shear modulus value during inspiration but also during expiration. This means that the best speed scores were achieved by athletes who had a higher RM shear modulus value during tidal inspiration and simultaneously a lower RM shear modulus value during tidal expiration. It may be that a better ability to relax the RMs allows for their greater contraction. When a muscle lengthens, the muscle spindle located inside the muscle is stretched, causing the muscle fibres to contract (*Bhattacharyya, 2017*). In turn, the comparable correlation values between each of the RMs and speed is probably due to the similar function of the DA and ICs. These muscles both affect chest movement (*Ratnovsky, Elad & Halpern, 2008*), produce axial rotations of the thorax (*Whitelaw et al., 1992*), and are important respiratory pump muscles (*Han et al., 1993*). Consequently, their work must be coordinated (*Han et al., 1993*). In addition, although the DA is the main RM, when the respiratory workload increases (high breathing efforts), the activity of ICs plays an important role (*Ratnovsky, Elad & Halpern, 2008*).

In view of the previous considerations, it is difficult to explain why footballers characterised by greater DA excursion and velocity during maximal inspiration had worse running scores. It was assumed that the increased stiffness of the DA during tidal breathing allowed greater stiffness of the DA during the Valsalva test because greater stiffness may result in lower DA excursion and velocity. Unfortunately, there are no studies connecting US assessment of RMs to speed in athletes, which greatly limits the interpretability of these preliminary findings.

### Endurance

Some studies have shown that exercises involving the RMs improve endurance by reducing energy demand (*Bahenský et al., 2021*) and increase aerobic tolerance (*Mackała et al., 2020*) in youth athletes. It has also been indicated that breathing technique can affect endurance through reduced respiratory work and delayed RM fatigue (*Bahenský et al., 2021*). This was the reason we hypothesised that endurance should be related to US of RMs in our study. This was not confirmed, as there was no relationship between the endurance and US parameters of RMs. In cited studies (*Bahenský et al., 2021*; *Mackała et al., 2020*), RMs strength was measured indirectly by analysing maximal inspiratory and expiratory pressure/forces. In the present study, for the first time, we have evaluated and related RMs with endurance directly by analysing US measurements (shear modulus, thickness, excursion, and velocity). An indirect method of assessing respiratory function is the result of many factors (including airway obstruction, respiratory compliance, and RM strength) that do not allow direct analysis of the RMs (*Pałac & Linek, 2022*). This may mean that the improvement in endurance in athletes is a more complex phenomenon unrelated to an exclusive change in RM morphology.

It is particularly surprising that there was no correlation between DA excursion and aerobic endurance in the present study. DA excursion is related to exercise capacity (*Shiraishi et al., 2020*) and can predict the improvement in exercise tolerance (*Shiraishi et al., 2020*) in patients (especially with problems related to the respiratory system). DA

excursion is related to pulmonary parameters like FVC, FEV1, and MIP, whereas DA velocity is related to FVC, MIP, and MEP (*Pałac & Linek, 2022*). All of these spirometry parameters are related to RM strength (*Pałac & Linek, 2022*). Thus, it was expected that greater DA excursion would predispose to better endurance in examined football players. Possibly in healthy people (and athletes who achieve higher performance in endurance tests than the non-athlete population), the DA excursion is not as important in order to improve endurance. An alternative explanation of the lack of correlation between DA excursion and aerobic endurance may be the relatively similar endurance (training) level of the footballers studied. However, there is a lack of scientific studies determining the significance of DA excursion in athletes. Hence, the present study results are difficult to interpret definitively.

## Limitations

Due to the small sample size, this study is of a preliminary nature. The study group consisted exclusively of football players from one team and age group, which may explain the high homogeneity of the participants' motor skills and US parameters. This, in turn, may have influenced the narrow dispersion of the variables and, ultimately, the correlation values. The results should not therefore be generalised to other sports. The participants were included in the analysis regardless of their position; studies have shown that footballers' profiles can vary according to where they play on the pitch (*Oliva-Lozano et al., 2020*). US examinations were performed only in the supine position. Another limitation was the collection of US measurements only during tidal breathing (except for excursion—maximal inspiration and tidal expiration). It seems necessary to include US assessment of the RMs during maximal respiratory efforts in future studies. For the purposes of this study, the athletes' endurance was indirectly determined. The MSRF is used as a test of aerobic capacity (*Voss & Sandercock, 2009*). The beep test can be used as a health indicator in children and adolescents (*Mayorga-Vega et al., 2016*), but it is a field test. Thus, the result should not be interpreted as a direct measurement of cardiorespiratory fitness, only as an estimation (*Mayorga-Vega et al., 2016*).

## Strength and implications

To date, RMs have never been directly investigated in the context of their association with athletes' performance. Although this is a pilot study, we have shown for the first time that some US parameters of the RMs may be related with motor skills (like speed in our study). From this perspective, we have confirmed that such exploration is justified. US provides an inexpensive and non-invasive tool for assessing RMs on wide populations. The methodology used in this report to assess RMs is easy accessible and reliable. Thus, it seems that the US of RMs in elite athletes is warranted in order to provide deeper insights into the role of RMs in the context of different motor abilities. Previous studies have confirmed the relationship between athletic performance and US parameters of lower-limb muscles (*Sarto et al., 2021*). It is also worth noting that RMs (mainly DA) function itself is related to pain sensation, stability, and balance. All these aspects are important in high-performance sport.

## CONCLUSIONS

Shear modulus of the RMs, DA excursion, and velocity are related to speed score in adolescent football players. In the examined population, endurance parameters were not related to any US parameters of RMs. The current state of knowledge does not allow us to conclusively determine how important US parameters of RMs can be in predicting performance parameters (for example endurance and speed) in young athletes. However, the results of the present study point to the need for further research into the role of US measurements of RMs in the development of motor skills.

### Funding

The study was fully funded by the Team of Biomedical Basis of Physiotherapy, The Jerzy Kukuczka Academy of Physical Education in Katowice. The funders had no role in study design, data collection and analysis, decision to publish, or preparation of the manuscript.

### Grant Disclosures

The following grant information was disclosed by the authors:
The Team of Biomedical Basis of Physiotherapy, The Jerzy Kukuczka Academy of Physical Education in Katowice.

### Competing Interests

The authors declare there are no competing interests.

### Author Contributions

- Małgorzata Pałac conceived and designed the experiments, performed the experiments, analyzed the data, prepared figures and/or tables, authored or reviewed drafts of the article, and approved the final draft.
- Damian Sikora performed the experiments, prepared figures and/or tables, authored or reviewed drafts of the article, and approved the final draft.
- Tomasz Wolny performed the experiments, authored or reviewed drafts of the article, and approved the final draft.
- Paweł Linek conceived and designed the experiments, performed the experiments, analyzed the data, prepared figures and/or tables, authored or reviewed drafts of the article, and approved the final draft.

### Human Ethics

The following information was supplied relating to ethical approvals (i.e., approving body and any reference numbers):

The study was approved by the Ethics Committee at the Jerzy Kukuczka Academy of Physical Education in Katowice

### Data Availability

The raw data is available in the Supplementary File.

## Supplemental Information

Supplemental information for this article can be found online at http://dx.doi.org/10.7717/peerj.15214#supplemental-information.

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
