# Peer review of "Relationship between respiratory muscles ultrasound parameters and running tests performance in adolescent football players. A pilot study"

_PeerJ, doi:10.7717/peerj.15214_

## Round 0.1 · original submission · Major Revisions

Some changes are needed to the article.

Georgian Badicu
Academic Editor
PeerJ Life & Environment

·

Basic reporting

Dear Authors,
I appreciate Your work on this interesting and inspiring study. I aim to give suggestions which can help to raise the quality of this work.
English is good, but in some places throughout the whole paper there are not adequate words used which need to be substituted by their more adequate synonyms – f.e. line 330 – “territorial test” – I’d rather write “field test”; “year group” should be replaced with “age group” – line 321, … Considering this I recommend to send the manuscript to a fluent English-speaking colleague who is an expert in the field of football, and in the field of spirometry and ask them to correct the linguistic aspect of the manus.

Experimental design

The study design and methods are well described.
I only can recommend to add more details fot the speed test, because there is no information about the: 1 starting position 2 distance of the starting line to the first line of the fotocells, 3 starting signal – was it present or no, 4 instruction to the players – were they instructed to hold the mrs after crossing the end line? 5 footwear that was used – was it equal for all the players ? if this was football boots or other type ?

Tables:
Table 1 – it is better to name it “experimental group characteristics” rather than study population which suggests a very big group
Replace “VO2max” with “calculated VO2max” unit should be (ml•kg-1•min-1)
Speed test – unit should be (s) rather than “sec”

Page 22 of the manus send to review – Table 2 “Relationship between ultrasound parameters and endurance test (multi-stage 20-m shuttle run) results” it should be “speed test”
Rather than “relationship” it should be written “correlations”.
Page 23 - Text over this table should be “speed test results” rather than “speed score results”
Page 24 – (table 3) – it should be “endurance test results” rather than “speed score results”

Consider using consistent therm for the endurance test results – in some places You write only “total” – table 3, in some places Total (no. of finished distances) ) – page 21 (additional “)” by the way; and in the methods section (line 180) – “MSRT ODC TOTAL” where the abbreviation ODC has not been explained at all.

Page 25 under the table – “Total - the total number of 20-meter distances run at each stage” – it is not true – it should be written “total number of completed 20-m repetitions” (during the whole test).

Validity of the findings

In my opinion, the information that the reader is missing is information about the position of the subjects on the pitch.
It is crucial to give this information, because the differences between goalkeepers and field positions are huge in the aspect of running performance. So putting GKs in one group with other field positions in this study implicates bias of the results. Secondly – there are differences in running performance between positions – defenders, midfielders and forwards, ref:
Oliva-Lozano JM, Fortes V, Krustrup P, Muyor JM. Acceleration and sprint profiles of professional male football players in relation to playing position. PLoS One. 2020 Aug 6;15(8):e0236959. doi: 10.1371/journal.pone.0236959. PMID: 32760122; PMCID: PMC7410317.
Owen, Adam & Djaoui, Léo & Newton, Matt & Malone, Shane & Mendes, Bruno. (2017). A contemporary multi-modal mechanical approach to training monitoring in elite professional soccer. Science and Medicine in Football. 1-6. 10.1080/24733938.2017.1334958.

As the Authors surely know, the endurance performance is affected by many factors, considering this it is crucial to search for correlation between field position, speed and endurance performance and RM parameters.
The study will be much more reliable if the authors show correlations of running speed and endurance with RM parameters in a group of similar field position or even divide the examined group on field positions and try to seek for correlations.

The second aspect – there are significant differences between players from the first squad and the players who play less – ref.:
Silva, Ana & Clemente, Filipe & Leão, César & Oliveira, Rafael & Badicu, Georgian & Nobari, Hadi & Poli, Luca & Carvutto, Roberto & Greco, Gianpiero & Fischetti, Francesco & Cataldi, Stefania. (2022). Physical Fitness Variations between Those Playing More and Those Playing Less Time in the Matches: A Case-Control Study in Youth Soccer Players. Children. 9. 1786. 10.3390/children9111786.

Considering this the examined players should be divided on 2 groups – those representing the first squad – and substitutes – and then seek for correlations.

This two factors may have vital impact on the results, and, at least, the readers should be informed about field position and 1-squad or substitute status of the examined players.

Last thing – the beep-test, even though it is widely used in football, it is not the best test to assess endurance. Having that in mind the Authors should consider changing the title of the paper.
Proposed title:
Are respiratory muscles’ shear modulus and thickness related to endurance and speed in
adolescent football players?

Correlations between respiratory muscles parameters assessed by USG and running performance in adolescent football players. A pilot study.

Additional comments

The paper is very interesting and already present good quality, but it still need major changes to be reliable.

·

Basic reporting

Of the 39 references, 18 are current and 21 have been published for more than five years. I suggest updating the theoretical framework.

Experimental design

Methods
It should present more clearly the design of the study. A CONSORT, or time line, should be presented in order to get a better view of the study design.
I suggest dividing the methodology into topics, design, sample, instruments, procedures and statistics.
The sample should be better explained with the number of subjects presented initially and then present the inclusion and exclusion criteria. Was any statistical calculation or program used to determine the sample size? Please mention. Is there a protocol regarding research in your lab? Please mention.
The instruments need to be better characterized, with manufacturer, city, state, if applicable, and country of manufacture, including the programs for data analysis used. Procedures should be better explained, such as the use of ultrasound.
Statistical treatment should be better detailed in order to better follow what has been done. It would be feasible to review the guidelines for Cohen (1988).

Validity of the findings

Results
Are presented satisfactorily. However, after being adapted to what is mentioned in the methodology, we believe that some presentations should be modified.
Discussion
Are presented satisfactorily. However, we suggest that you begin the discussion by briefly reaffirming the objectives.
It should reaffirm the objectives and start discussing the results in the chronological order that appear in the item results.
We also suggest that what was mentioned at the beginning of the discussion be placed at the end of it, as the statements found in the first paragraph would be a justification that could precede the limitations of the study.
Conclusion
Are presented satisfactorily. However, in addition to what was mentioned, it should bring some practical applications of the findings, which does not occur.

Additional comments

Title
The title in the form of a question did not clearly present what was done in the manuscript. I suggest that it be redone and placed in the affirmative.
Abstract
It is written in a structured way, however, the methodology is written in a very summarized way which ends up making the findings and conclusions of the article.
The methodology has to be better explained in the methodology.
In the results, absolute and statistical values must be presented to facilitate the understanding of what was done in the study.
Please confirm that the keywords are presented as described in health sciences.
Introduction
The introduction is satisfactorily well written, moving from general to specific.
However, it should initially present a more general approach and gradually address the problem (gap) and then present the objective.
The problem is not well identified, the fact that there are few studies would not be a sustainable problem. There must be a better explanation to support the study.
Goals must come before hypotheses.

---

## Round 0.2 · Minor Revisions

Authors, The original Academic Editor is no longer available and so I have taken over the handling of your submission.

There are additional minor changes (edits and comments) required throughout the manuscript before I can recommend it to be published. See attached edits and comments Thanks,

·

Basic reporting

Dear Authors and Editor - I appreciate how much work You did to increase the quality of the paper. I am happy that my review could help You.
At this stage - all my previous suggestions have been corrected. I really appreciate also the work of the second reviewer and the Authors' reaction to his suggestions.

One last thing need imporveing - and this is still the title ;)

rather than this version:
"Relationship between respiratory muscles ultrasound measurements and running tests in adolescent football players"

I would suggest considering the below version:

Relationship between respiratory muscles ultrasound parameters and running tests performance in adolescent football players. A pilot study.

Experimental design

In future I really recommend excluding goalkeepers from the football players groups because the characteristics of GK is very different.

Methods - in short runs suing photocells there is a practice to put the toes of the leading leg 5-10 (depends on different protocols) before tha line of the photocells - it helps the participants not to cross the beam of the photocell with their moving arm before they really start.

Validity of the findings

my previous suggestions were corrected

Additional comments

All the best to the Editor, second reviewer and the Authors.

·

Basic reporting

After reading the revised manuscript, and evaluating point by point the adaptations presented by the author. And after evaluating the answers presented. We consider that the manuscript is ready to be published, considering that the adjustments and/or responses met our expectations and answered our questions.

Experimental design

After reading the revised manuscript, and evaluating point by point the adaptations presented by the author. And after evaluating the answers presented. We consider that the manuscript is ready to be published, considering that the adjustments and/or responses met our expectations and answered our questions.
The drawing is correct

Validity of the findings

Meets the needs of the study

Additional comments

I believe that the manuscript is ready to be published, considering that the adjustments and/or responses met our expectations and answered our questions. It has an appropriate design, meets the scientific criteria for publication in the form of an article.

---

## Round 0.3 · accepted · Accept

Dr Linek and co-authors have made all of my recommended edits/comments. I therefore recommend the manuscript to be published. Thank you for supporting PeerJ, we look forward to future manuscript submissions from your research team. Thanks, A/Prof Mike Climstein